# Precision and accuracy of FEV$_1$ measurements from the Vitalograph copd-6 mini-spirometer in a healthy Ugandan population

Wajd Abbas Hassan Hansen[1,2], Vivi Schlünssen[2,3], Erik Jørs[4,5], Daniel Sekabojja[6], John C. Ssempebwa[7], Ruth Mubeezi[7], Philipp Staudacher[8,9], Samuel Fuhrimann[10], Martin Rune Hassan Hansen[2,3]*

1 Lægerne i Hirtshals APS, Hirtshals, Denmark, 2 Research Unit for Environment, Work and Health, Danish Ramazzini Center, Department of Public Health, Aarhus University, Aarhus, Denmark, 3 National Research Center for the Working Environment, Copenhagen, Denmark, 4 Department of Occupational and Environmental Medicine, Odense University Hospital, Odense, Denmark, 5 Occupational and Environmental Medicine, Department of Clinical Research, University of Southern Denmark, Odense, Denmark, 6 Uganda National Association of Community and Occupational Health, Kampala, Uganda, 7 School of Public Health, Makerere University, Kampala, Uganda, 8 Eawag, Swiss Federal Institute of Aquatic Science and Technology, Dübendorf, Switzerland, 9 Institute of Biogeochemistry and Pollutant Dynamics, ETH Zurich, Zurich, Switzerland, 10 Institute for Risk Assessment Sciences, Utrecht University, Utrecht, Netherlands

* martinrunehassanhansen@ph.au.dk

**Data Availability Statement:** The minimal dataset (S4 File) contains the raw data for both the participant spirometries and the copd-6 calibration

## Abstract

### Objective

Evaluate the accuracy and precision of the copd-6 mini-spirometer for FEV$_1$ in a rural Ugandan population.

### Methods

In a cross-sectional study, 171 smallholder farmers performed spirometry with copd-6, and a diagnostic-quality spirometer.

### Results and discussion

The copd-6 underestimated FEV$_1$ at low flows and overestimated FEV$_1$ at high flows. Across all participants, the device slightly overestimated FEV$_1$ by 0.04 [0.02; 0.06] L. Calibration data showed similar patterns.

### Conclusion

The copd-6 could be considered as an affordable tool for research on lung function impairment in resource-constrained settings. However, further validation in a study population with obstructive lung disease is needed.

checks. To protect participant confidentiality, data on demographics and family relationships have not been included in the file. For the subset of participants that consented to data sharing, a deidentified dataset including demographics and family relationships is available on reasonable request. For access, please contact the Department of Public Health at Aarhus University at ph@au.dk. Access requires approval from the MakSPH-HDREC and the Danish Data Protection Agency.

**Funding:** For this work, MRHH received funding from Aarhus University Research Foundation (project number 81231, https://auff.au.dk/en/) and the National Research Center for the Working Environment (project number 10322, https://nfa.dk/). The funders had no role in study design, data collection and analysis, decision to publish, or preparation of the manuscript. The project was not supported (financially or otherwise) by Vitalograph.

**Competing interests:** The authors have declared that no competing interests exist.

**Abbreviations:** ATS, American Thoracic Society; AUC, Area Under the Curve; COPD, chronic obstructive pulmonary disease; $FEV_1$, forced expiratory volume in 1 second; $FEV_6$, forced expiratory volume in 6 seconds; FVC, forced vital capacity; IQR, interquartile range; MakSPH-HDREC, Higher Degrees Research and Ethics Committee at Makerere University School of Public Health, Kampala, Uganda; PEXADU, Pesticide Exposure, Asthma and Diabetes in Uganda" (project title); UNCST, Uganda National Council for Science and Technology.

# Introduction

Chronic obstructive pulmonary disease (COPD) causes over 3 million deaths annually; more than 80% in low and middle-income countries [1]. In Uganda, the rural prevalence of COPD among adults > 35 years is 6.1% [2]. Spirometry is necessary for diagnosis, yet often unavailable or prohibitively expensive [3]—a typical situation in many low- and middle-income countries [3].

The copd-6 mini-spirometer (Vitalograph, Ennis, Ireland) is relatively inexpensive, costing around 125 USD–compared to more advanced spirometers that often cost upward of 600 USD. The copd-6 has been shown to have reasonable precision for the diagnosis of COPD [4], and holds promise as a diagnostic tool in resource-constrained settings. But the accuracy of the device remains insufficiently described, limiting its potential for use by local researchers investigating risk factors for lung function impairment, since inaccurate measurements have the potential to bias exposure-response relationships. The purpose of this study is therefore to evaluate both the accuracy and precision of the copd-6 mini-spirometer for the measurement of $FEV_1$ in a Ugandan population.

# Methods

## Study population

Data was collected as a part of the "Pesticide Exposure, Asthma and Diabetes in Uganda" (PEXADU) project, which was a study on the possible associations between pesticide exposure, pulmonary function and diabetes in a cohort of smallholder farmers in Uganda [5–7]. Details regarding participant recruitment and inclusion/exclusion have been presented elsewhere [5–7]. In brief, we recruited 364 farmers from the Wakiso District in central Uganda with the help of two local farmer's organizations. We visited a number of smaller farmer's groups affiliated with these two organizations and invited all members aged > 18 years to participate, except pregnant women. Participants came to the project examination center at baseline in September-October 2018, with two rounds of follow-up in November-December 2018 and January-February 2019, respectively. All data analyzed in this paper are from the baseline examination.

## Spirometry testing

Out of 364 participants, 304 performed spirometry. The majority of the remaining participants were excluded because they self-reported one of the following: Myocardial infarction in the last 3 months, angina pectoris, hemoptysis, any surgery in the last 3 months, aortic aneurism, history of pulmonary embolism, active tuberculosis or other current respiratory infection. We also excluded one individual with severe hypertension (defined as blood pressure > 200 mmHg systolic or > 120 mmHg diastolic).

Participants underwent spirometry with both a diagnostic-quality spirometer (MicroDL, Micro Medical, Rochester, Kent, England) and a copd-6 mini-spirometer (lot number 0317/2018). Tests were conducted between 7 AM and 5 PM. A pseudo-random number generator randomized the order of the devices in the lung function test. Only a few minutes passed between testing with each of the two devices. For the MicroDL device, participants first blew five times into the device. If their results did not fulfill standard quality criteria [8], they got four additional attempts. To avoid fatigue, participants only blew three times with the copd-6.

## Ethics approval and consent to participate

The project was conducted in accordance with the Declaration of Helsinki. All participants gave written informed consent before inclusion and were financially compensated for lost

earnings on the examination day. Ethical approval was granted by the Higher Degrees Research and Ethics Committee at Makerere University School of Public Health, Kampala, Uganda (MakSPH-HDREC, registration number 577) and the Uganda National Council for Science and Technology, Kampala, Uganda (registration number HS234ES).

### Calibration checks

The calibration of the copd-6 devices was checked daily using a 3-liter calibration syringe (MIR 919000, Medical International Research Inc., Rome, Italy). The syringe was emptied three times with each of three speeds: Slow (as slowly as possible, while finishing within 6 seconds), medium, and fast (as fast as possible). The calibration of the MicroDL devices was checked in a similar way, except that there was no time limit for the slow plunges of the calibration syringe. Only copd-6 calibration check data will be presented here. Calibration check data for the MicroDL devices have been reported elsewhere [6].

### Data entry

Data from the MicroDL were extracted in digital format using Spida 5 PC software (MicroDL, Micro Medical, Rochester, Kent, England). Data from the copd-6 were entered directly in a structured database at the time of testing or calibration, using the ODK Collect app [9].

### Quality control

The copd-6 device gives a warning in case the subject coughs during a blow, or in case of slow starts, and such blows were excluded. We excluded all copd-6 results from participants with less than two accepted blows. Furthermore, we excluded non-repeatable copd-6 results, defined as a difference in the best and second-best $FEV_1$ and $FEV_6 > 0.25$ liters.

A medical doctor assessed the quality of MicroDL spirometry according to modified ATS criteria [8], as previously reported elsewhere [6, 7]. All MicroDL results were excluded if the participant had performed less than two acceptable blows, or if the results were non-repeatable, defined as a difference in the best and second-best $FEV_1$ or $FVC > 0.25$ liters.

### Statistical analyses

Participant $FEV_1$ was analyzed in Bland-Altman plots. Trends in $d = (FEV_{1,copd6} - FEV_{1,MicroDL})$ as a function of $m = (FEV_{1,copd6} + FEV_{1,MicroDL})/2$ was analyzed in a mixed effect model with a fixed effect for $m$ and a random effect for participant family. To account for non-linearity, $m$ was modelled using restricted cubic splines with four knots. Since the MicroDL reported FVC and not $FEV_6$, analyses of $FEV_6$ and $FEV_1/FEV_6$ were limited to 21 participants whose MicroDL spirograms showed that their $FEV_6$ and FVC were equal. For copd-6 calibration check data, reported $FEV_6$ vs. speed of pushing the piston was analyzed using Spearman's rank correlation.

Data management and analyses were performed in Stata 15 (StataCorp, College Station, Texas, United States). The statistical analyses were specified a priori and the analysis protocol published in an online repository before analysis [10]. All deviations between the protocol and the final analyses are listed in S1 File.

## Results

Out of 364 participants in the PEXADU study, 304 (84%) performed spirometry, and 171 (47%) fulfilled quality criteria for both the MicroDL and copd-6 devices and were included in

**Table 1. Demographic characteristic of the population.**

|  | All participants | Performed spirometry | Included in analyses |
|---|---|---|---|
| n | 364 | 304 | 171 |
| Sex |  |  |  |
| Female, n (%) | 250 (68.7) | 206 (67.8) | 107 (62.6) |
| Male, n (%) | 114 (31.3) | 98 (32.2) | 64 (37.4) |
| Age, years (IQR) | 46.6 (36.7; 56.5) | 45.3 (35.6; 54.7) | 46.0 (36.9; 55.8) |
| Years of full-time education (IQR) | 7.0 (5.0; 11.0) | 7.5 (6.0; 11.0) | 7.0 (6.0; 11.0) |
| BMI, kg/m$^2$ (IQR) | 23.3 (21.1; 26.8) | 23.3 (21.2; 26.6) | 23.3 (21.1; 26.9) |
| Height, cm (IQR) | 158.9 (154.1; 164.3) | 159.1 (154.3; 164.3) | 159.1 (153.8; 165.0) |
| Ever-smoker |  |  |  |
| No, n (%) | 322 (88.5) | 274 (90.1) | 148 (86.5) |
| Yes, n (%) | 42 (11.5) | 30 (9.9) | 23 (13.5) |
| Pack-years for ever-smokers (IQR) | 2.1 (0.9; 6.8) | 1.8 (0.7; 6.8) | 1.3 (0.6; 6.8) |
| Cooking fuel type in household |  |  |  |
| Charcoal, n (%) | 61 (16.8) | 55 (18.1) | 29 (17.0) |
| Wood, n (%) | 298 (81.9) | 244 (80.3) | 138 (80.7) |
| No food cooked in household, n (%) | 1 (0.3) | 1 (0.3) | 1 (0.6) |
| Other, n (%) | 4 (1.1) | 4 (1.3) | 3 (1.8) |

Continuous variables presented as median (interquartile range, IQR).

the analyses. Demographic information on the study population is shown in Table 1. An overview of participant inclusion is provided in Fig 1.

The mean $FEV_1$ from the MicroDL was 2.48 L (Table 2). Overall, the copd-6 slightly overestimated $FEV_1$, with a mean difference (defined as $FEV_{1,copd6}$ –$FEV_{1,MicroDL}$) of 0.04 [0.02; 0.06] L. Most participants had an absolute difference in $FEV_1 < 0.25$ L between the two devices, although there were some outliers (Fig 2). Across participants, the average of participants' $FEV_1$ from the copd-6 and MicroDL devices ($m$) ranged from 0.98 L to 4.68 L. Based on the trend in the mixed effect model (i.e., not individual participants), the difference between the two devices was -0.06 [-0.17; 0.05] L at the minimum $m$ (0.98 L), and 0.21 [0.08; 0.33] L at the maximum $m$ (4.68 L). We found similar results in a number of sensitivity analyses that were stratified by the specific devices used for testing, stratified by the order of testing (copd-6 before MicroDL or vice versa), only included the first three blows with the MicroDL, or used stricter repeatability criteria (S2 File).

For the 21 subjects where $FEV_6$ could be directly compared between the copd-6 and MicroDL, the overall difference in mean $FEV_1/FEV_6$ was -0.02 [-0.03; 0.00]. The difference seemed to depend on the value of $FEV_1/FEV_6$; the copd-6 overestimated ratios close to 1 and underestimated lower ratios (S2 File).

Calibration data showed the same pattern: Underestimation of volumes at low flow and overestimation at high flow (Fig 3), but most results for low and medium flow were within 3.00 liters ± 3% (the accuracy required by ATS guidelines) [8].

## Discussion

Overall, the copd-6 measures $FEV_1$ reasonably well in our relatively healthy study population. However, the device slightly underestimates volume at low flows and overestimates volume at high flows. In addition, it may underestimate $FEV_1/FEV_6$, but as $FEV_1/FEV_6$ could be analyzed for only few subjects, we have limited confidence in this latter finding.

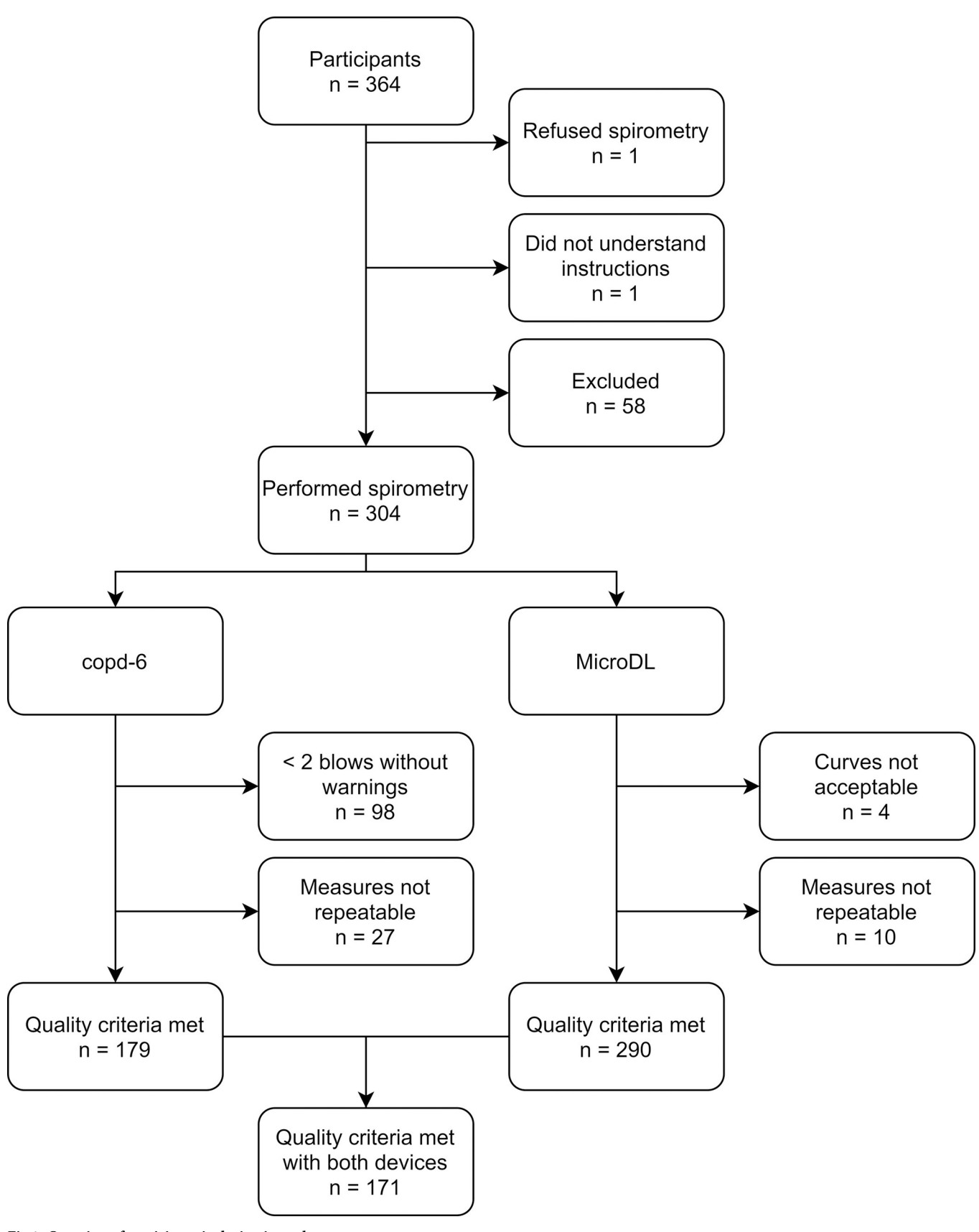

**Fig 1. Overview of participant inclusion in analyses.**

**Table 2. Summary metrics for results from MicroDL and copd-6 spirometers.**

| | | MicroDL | copd-6 | | |
| --- | --- | --- | --- | --- | --- |
| | | | Before MicroDL | After MicroDL | Total |
| **Number of participants** | | 171 | 84 | 87 | 171 |
| **$FEV_1$ (L)** | Median [2.5 percentile; 97.5 percentile] | 2.37 [1.46; 3.87] | 2.33 [1.40; 3.89] | 2.39 [1.22; 3.94] | 2.38 [1.22; 3.94] |
| | Mean [95% CI] | 2.44 [2.34; 2.54] | 2.48 [2.32; 2.63] | 2.48 [2.33; 2.63] | 2.48 [2.37; 2.59] |
| **$FEV_6$ (L)** | Median [2.5 percentile; 97.5 percentile] | N/A | 2.81 [1.47; 4.46] | 2.77 [1.30; 4.59] | 2.77 [1.30; 4.59] |
| | Mean [95% CI] | N/A | 2.83 [2.65; 3.01] | 2.82 [2.65; 3.00] | 2.82 [2.70; 2.95] |
| **FVC (L)** | Median [2.5 percentile; 97.5 percentile] | 2.89 [1.63; 4.99] | N/A | N/A | N/A |
| | Mean [95% CI] | 2.96 [2.84; 3.09] | N/A | N/A | N/A |

A recent systematic review of 14 original papers found that measurements from the copd-6 had an Area Under the Curve (AUC) of 0.90 for the diagnosis of COPD [4]. It should be noted that AUC reflects only precision, and not accuracy. As the number of persons with airway obstruction in our study was low, we do not have statistical power to calculate sensitivity and specificity of the copd-6 for obstruction; hence, our results cannot be directly compared with the systematic review. However, some authors of original studies have reported numerical

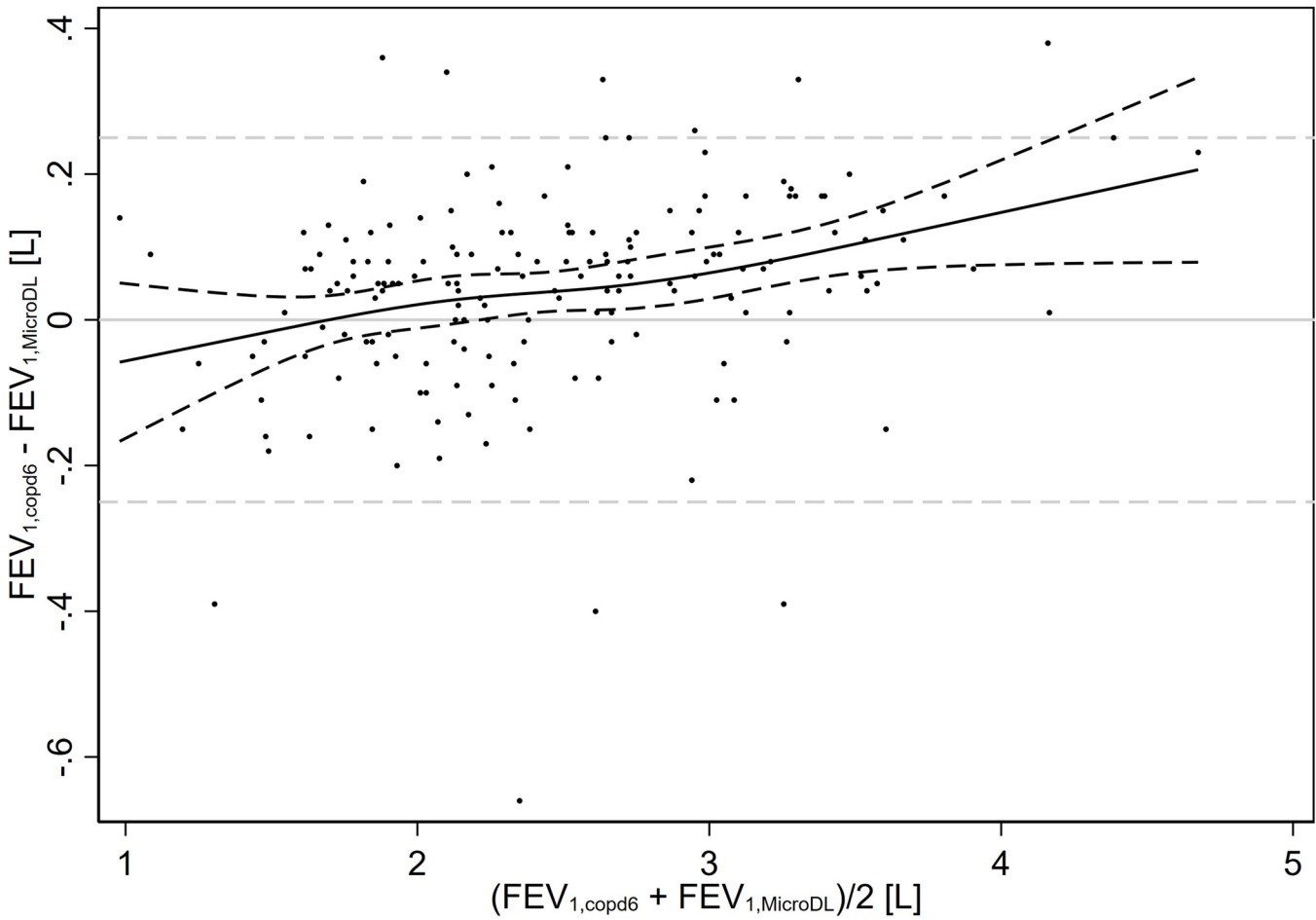

**Fig 2. Bland-Altman plot of copd-6 and MicroDL participant results for $FEV_1$.** Each dot represents one participant. Solid gray line = 0.00 L. Dashed gray lines = ± 0.25 L. Black lines = trend with 95% confidence interval.

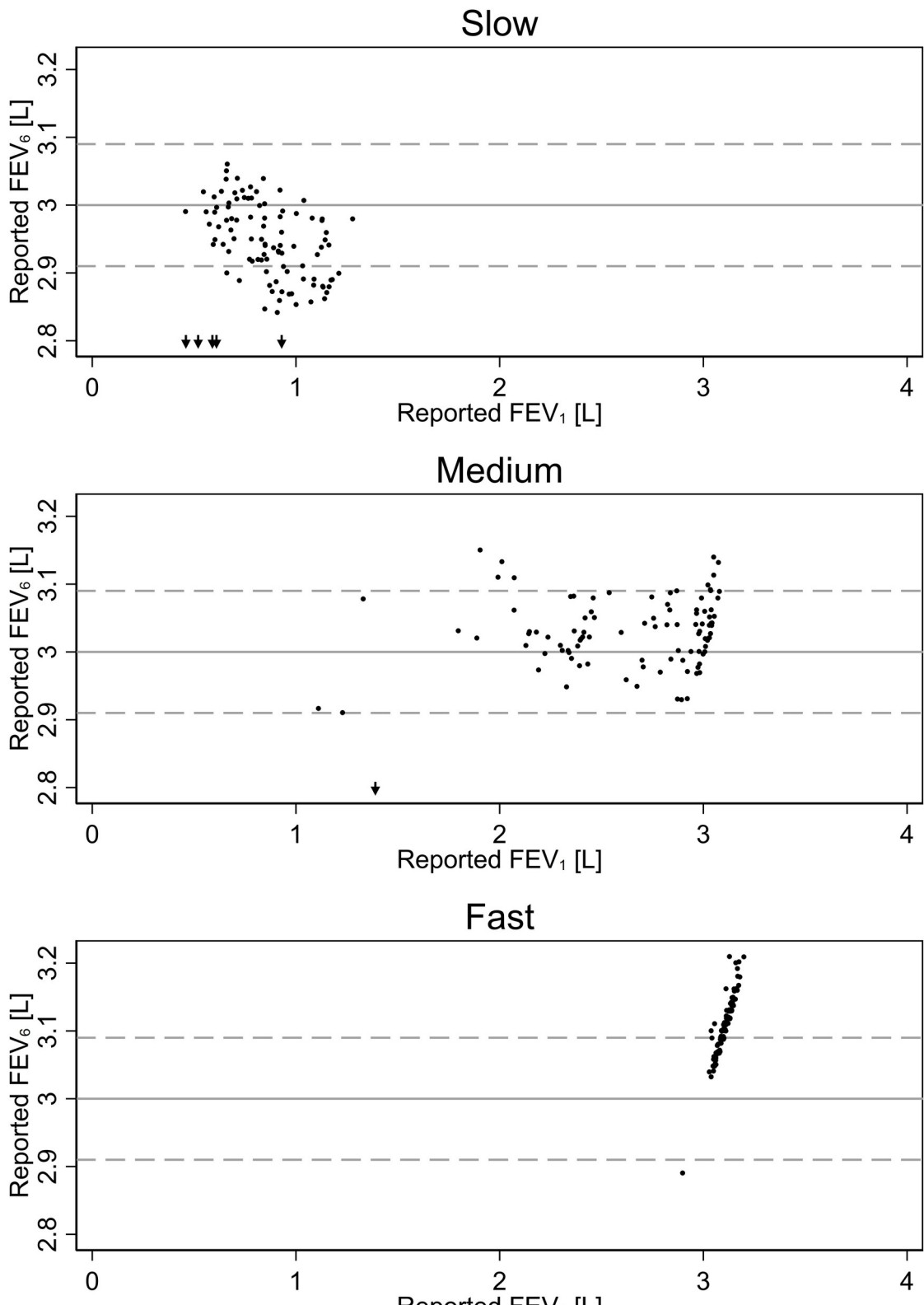

**Fig 3. Scatterplots of calibration check data for copd-6, stratified by the speed of pushing the piston of the calibration syringe.** Spearman's $\varrho = 0.82$, p < 0.001. Each dot represents one plunge of the calibration syringe. Arrows pointing downward are plunges with reported $FEV_6$ < 2.8 liters. Gray lines located at 3.00 liters ± 3%.

results from the copd-6 that allow direct comparisons. In accordance with our results, one study showed a small but statistically significant overestimation of $FEV_1$ by the copd-6, with underestimation at low flows and overestimation at high flows [11]. But on the other hand, seven studies indicated that the copd-6 underestimated $FEV_1$ [12–18], with estimated differences from diagnostic-quality spirometers ranging from 0.01 to 0.17 L. In two of the latter studies, Bland-Altman plots indicated that the copd-6 underestimates $FEV_1$ more when the $FEV_1$ is high [12, 18], while one study did not show a clear trend in $FEV_1$ bias for the copd-6 [16]. We do not have a clear explanation for the differences between previous studies and the present one. Since we saw the same trends in participant and calibration check data, our results are not explained by any insufficiently calibrated MicroDL devices. Given that calibration check data from two different copd-6 devices gave very similar results (S3 File), our results are also not explained by a defective copd-6. A possible explanation for the between-study discrepancies could be the use of copd-6 devices from different lots that therefore differed in calibration, but that is purely speculative.

Out of the 304 participants who underwent spirometry in the current study, only 179 (59%) fulfilled quality criteria with the copd-6, while 290 (95%) fulfilled quality criteria with the MicroDL device. Since the same technicians were responsible for testing with the copd-6 and MicroDL devices, this difference is unlikely to be due to problems with coaching, as poor coaching would have influenced both copd-6 and MicroDL results. Furthermore, the corresponding author supervised the spirometry technicians during data collection, and it was his impression that the technicians vigorously coached the participants during each blow. Out of the 125 persons whose copd-6 results were excluded, 98 (78%) were excluded because they had performed less than two blows without warnings. It could be speculated that the copd-6 is perhaps too conservative when assessing the quality of individual blows (and labelling them as "unacceptable" by giving a warning), but since the device does not export spirograms for manual review, we cannot investigate this further.

A number of sensitivity analyses have been performed to investigate the robustness of our findings. Participants blew up to twelve times into the copd-6 and MicroDL devices, which could theoretically have biased our results due to fatigue after repeated forced exhalations. However, the order of testing (copd-6 before MicroDL or vice versa) was randomized, and sensitivity analyses stratified by order of testing showed similar results as the main analysis (S2 File), meaning that our findings are unlikely to be considerably influenced by patient fatigue. We also do not think that the different number of blows allowed with the two devices (three blows with the copd-6, up to nine blows with the MicroDL) poses a problem, as a sensitivity analysis limited to the first three blows with the MicroDL gave similar results as the main analysis (S2 File). To maximize the amount of available data and avoid selection bias, our main analysis used less strict criteria for the reproducibility of $FEV_1$, FVC and $FEV_6$ than recommended by the ATS [8]. However, while using less strict reproducibility criteria is expected to introduce imprecision in our estimates, there is no reason to think that it would lead to bias, and a sensitivity analysis using the official ATS criteria [8] also gave similar results as the main analysis (S2 File).

Study participants were recruited in a non-random manner by inviting members of a number of farmer's groups in one specific district of Uganda, meaning that the study population may not be representative of Ugandan smallholder farmers overall. However, as the goal of the present study was not to assess the prevalence of lung function impairment among Ugandan smallholder farmers in general, but rather to compare $FEV_1$ measurements made with the copd-6 and MicroDL spirometers in a specific study population, we do not think that this poses a threat to the validity of our findings regarding $FEV_1$. As participants were relatively healthy, and the number of participants with airway obstruction was limited, it was difficult

for us to assess the validity of $FEV_1/FEV_6$ measured by the copd-6. To further investigate the validity of the $FEV_6$ and $FEV_1/FEV_6$ measurements, a new study would be needed, with a larger proportion of participants with airway obstruction, and results from the copd-6 should be compared to a diagnostic-quality spirometer that reports $FEV_6$ in addition to FVC.

The tendency of the copd-6 to slightly underestimate volume at low flows and overestimate volume at high flows means that if it is used to measure $FEV_1$ in studies of risk factors for pulmonary function impairment, exposure-response relationships may be biased away from the null. Researchers using $FEV_1$ measurements from the copd-6 need to check the calibration of their devices and account for any inaccuracies.

## Conclusion

Overall, in our relatively health study population the copd-6 slightly overestimates $FEV_1$ by 0.04 L, but the bias is flow-dependent. The copd-6 could be considered as an affordable way to conduct research on pulmonary function impairment in resource-constrained settings, but careful calibration checks are necessary. Further validation in a study population with obstructive diseases is needed, and the copd-6 should be compared to a gold standard spirometer that reports $FEV_6$ in addition to FVC.

## Supporting information

**S1 File. Deviations between analysis protocol and final analyses.**
(PDF)

**S2 File. Supplementary results for participant data.**
(PDF)

**S3 File. Supplementary results for copd-6 calibration check data.**
(PDF)

**S4 File. Minimal dataset.**
(XLSX)

**S5 File. STROBE checklist.**
(PDF)

## Acknowledgments

We wish to thank the participants and our collaborators from the Diálogos Foundation, the Uganda National Association of Community and Occupational Health, the Agency For Integrated Rural Development, Wakiso District Farmers Association and Caritas Uganda. We are thankful for the efforts of the field team: Amusa Wamawobe, Betty Kateregga, Brenda Wagaba, Evans Twin, Grace Lubega, Jonathan Mugweri, Imelda Namatovu, Joviah Gonza, Lydia Yariwo and Timothy Masaba.

Parts of Tables 1 and 2 have previously been published elsewhere and are reproduced here with permission [7].

## Author Contributions

**Conceptualization:** Vivi Schlünssen, Erik Jørs, Daniel Sekabojja, John C. Ssempebwa, Ruth Mubeezi, Philipp Staudacher, Samuel Fuhrimann, Martin Rune Hassan Hansen.

**Data curation:** Martin Rune Hassan Hansen.

**Formal analysis:** Wajd Abbas Hassan Hansen, Martin Rune Hassan Hansen.

**Funding acquisition:** Vivi Schlünssen, Martin Rune Hassan Hansen.

**Investigation:** Martin Rune Hassan Hansen.

**Methodology:** Vivi Schlünssen, Erik Jørs, Daniel Sekabojja, John C. Ssempebwa, Ruth Mubeezi, Philipp Staudacher, Samuel Fuhrimann, Martin Rune Hassan Hansen.

**Project administration:** Martin Rune Hassan Hansen.

**Resources:** Daniel Sekabojja.

**Supervision:** Vivi Schlünssen, Erik Jørs.

**Writing – original draft:** Wajd Abbas Hassan Hansen, Martin Rune Hassan Hansen.

**Writing – review & editing:** Vivi Schlünssen, Erik Jørs, Daniel Sekabojja, John C. Ssempebwa, Ruth Mubeezi, Philipp Staudacher, Samuel Fuhrimann.

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
