## [Decision Letter · Decision Letter 0]

29 Oct 2020

PONE-D-20-31069

The Vitalograph copd-6 mini-spirometer as more than a screening device: Validation of FEV_1_ in a healthy Ugandan population

PLOS ONE

Dear Dr. Hansen,

Thank you for submitting your manuscript to PLOS ONE. After careful consideration, we feel that it has merit but does not fully meet PLOS ONE’s publication criteria as it currently stands. Therefore, we invite you to submit a revised version of the manuscript that addresses the points raised during the review process.

We look forward to receiving your revised manuscript.

Kind regards,

Davor Plavec, MD, MSc, PhD, Prof.

Academic Editor

PLOS ONE

Additional Editor Comments:

Please make revisions or write a detailed arguments according to reviewers' suggestions.

Journal Requirements:

2. In your Methods section, please provide additional information about the participant recruitment method and the demographic details of your participants. Please ensure you have provided sufficient details to replicate the analyses such as: a) the recruitment date range (month and year), b) a description of any inclusion/exclusion criteria that were applied to participant recruitment, c) a table of relevant demographic details, d) a statement as to whether your sample can be considered representative of a larger population, e) a description of how participants were recruited, and f) descriptions of where participants were recruited and where the research took place.

3. We noted in your submission details that a portion of your manuscript may have been presented or published elsewhere.

"Summary statistics for demographic variables and for MicroDL spirometry results have been included in two previous papers that have been uploaded as "Related papers". These two papers focused on health effects of pesticides in the same study population. Since we have not previously published any results for the copd-6 mini-spirometer, and since the demographic information in table 1 is included only to provide context for the main analyses related to the performance of the spirometers, we do not find that this constitutes dual publication."

Reviewers' comments:

Reviewer's Responses to Questions

**Comments to the Author**

1. Is the manuscript technically sound, and do the data support the conclusions?

Reviewer #1: Yes

Reviewer #2: Yes

2. Has the statistical analysis been performed appropriately and rigorously? 

Reviewer #1: Yes

Reviewer #2: I Don't Know

3. Have the authors made all data underlying the findings in their manuscript fully available?

Reviewer #1: Yes

Reviewer #2: Yes

4. Is the manuscript presented in an intelligible fashion and written in standard English?

Reviewer #1: Yes

Reviewer #2: Yes

5. Review Comments to the Author

Reviewer #1: Authors recruited well defined investigated population, and applied all exclusion criteria  as needed according to guidelines before spirometry measurements. Their results are the big step about accuracy of copd 6 as a device for ventilatory lung function. Result of

 little difference between the two tested devices for spiromery between 0.06 and 0.21 L for minimal and maximal FEV1 measurements is valuable and worth of attention. Also, very interesting results is, from both analysis of the same FEV6 and FVC measured on both devices as well as from calibration data, that volumes are underestimated at low flows, and overestimated at high flows. It is very important finding, because screening on copd-6 device could overdiagnosed obstructive ventilatory disorders. They convenience us that copd 6 could be a good and cheaper alternative for a screening of obstructive lung diseases, if all users are aware of limitations.

Considering methods of performing spirometry I have some objections. Investigation team stated 5  efforts on a diagnostic-quality spirometry, and eventually 4 additional blew if participants did not fulfill standard quality criteria, while according to the ATS/ERS guideline it should be at least three, and most 8 maneuvers for acceptable spirometry. On the copd-6 spirometer they measured just three maneuvers, what is an obstacle, because fatigue is expected after too many efforts of forces expiration. Here were 12 blew into devises in some patients, what it much, and put in question quality of results. Time frame between two spirometric measurements is not stated on a diagnostic-quality spirometer and on a copd-6 spirometer, what is another question.

Also, due to diurnal variation of bronchial muscle tone, it is necessary to put the information of time during the day when the measurements were performed.

Again, considering cut-off value for non repeatable spirometric result between two best FEV1 and FVC measurements, the chosen value of difference < 0.25 L is too high, due to the  ATS/ERS guideline, the cut-off for difference should be < 0.15 L.

Another unusual results is that in healthy, rather young and not obese population (46.6 years and BMI 23.3), only 47% of the investigated subjects fulfilled quality criteria for both spirometry devices. Does it mean that the personal was not enough motivated for the participants, or why – this fact need further explanation.

Reviewer #2: The high drop out in the COPD6 group should be explained in more details and elaborated in the discussion and the conclusion. The fact that the investigation was done in the population without obstruction should be more accentuated in the conclusion.

106- less than two or <2 not mixed

6. PLOS authors have the option to publish the peer review history of their article (what does this mean?). If published, this will include your full peer review and any attached files.

Reviewer #1: **Yes: **Sanja Popović-Grle, M.D., Ph.D., professor of the School of Medicine University of Zagreb

Reviewer #2: No

---

## [Author Response · Author response to Decision Letter 0]

9 Jan 2021

Journal requirements

Comment

Response

We confirm that the manuscript meets the style requirements, including those for file naming.

Comment

2. In your Methods section, please provide additional information about the participant recruitment method and the demographic details of your participants. Please ensure you have provided sufficient details to replicate the analyses such as: a) the recruitment date range (month and year),

Response

More details on recruitment methods and the demographic details are now added to the section “Methods” � “Study population”. All data presented in the current paper are from the baseline examinations of the PEXADU study that were conducted in September-October 2018 (see https://doi.org/10.1136/oemed-2020-106439).

Comment

b) a description of any inclusion/exclusion criteria that were applied to participant recruitment,

Response

We have now added this information in the section “Methods” � “Study population”. We recruited participants from a number of farmer’s groups, and invited all members of the visited farmer’s groups, except persons aged < 18 and pregnant women.

 

Comment

c) a table of relevant demographic details,

Response

Demographic information is provided in Table 1.

Comment

d) a statement as to whether your sample can be considered representative of a larger population,

Response

We have added a paragraph in the “Discussion” section where we discuss any problems with external validity. Study participants were recruited in a non-random manner by inviting members of a number of farmer's groups in one specific district of Uganda, meaning that the study population may not be representative of Ugandan smallholder farmers overall. However, as the goal of the present study was not to assess the prevalence of lung function impairment among Ugandan smallholder farmers in general, but rather to compare FEV1 measurements made with the copd-6 and MicroDL spirometers in a specific study population, we do not think that this poses a threat to the validity of our findings.

Comment

e) a description of how participants were recruited, and

Response

A description of the recruitment has been added to the section “Methods” � “Study population”. The section includes a reference to a previous paper from the same study population (https://doi.org/10.1136/oemed-2020-106439), where additional details on inclusion/exclusion are provided.

Comment

f) descriptions of where participants were recruited and where the research took place.

Response

The section “Methods” � “Study population” describes that the project took place in the Wakiso District of Uganda. As described above, we have now added that participants were recruited by attending meetings held by a number of farmer’s groups belonging to two local farmer’s organizations in the district. As indicated in the section, participants came to a project examination center to participate.

 

Comment

3. We noted in your submission details that a portion of your manuscript may have been presented or published elsewhere.

"Summary statistics for demographic variables and for MicroDL spirometry results have been included in two previous papers that have been uploaded as "Related papers". These two papers focused on health effects of pesticides in the same study population. Since we have not previously published any results for the copd-6 mini-spirometer, and since the demographic information in table 1 is included only to provide context for the main analyses related to the performance of the spirometers, we do not find that this constitutes dual publication."

Response

The previous paper on MicroDL results has not yet been published, but has been accepted for publication in Thorax (https://thorax.bmj.com/). Some descriptive statistics for demographic variables have also been included in a peer-reviewed paper published on https://doi.org/10.1136/oemed-2020-106439 (but that paper did not include any spirometry results). Furthermore, the summary statistics for the MicroDL that are presented in the second column of Table 2 have also been included in the paper submitted to Thorax. If you prefer, we can remove Table 1 and the second column of Table 2 from the paper. However, we find that without these purely descriptive statistics, readers will have trouble putting the findings of the current paper into proper context. Again, we would like to emphasize that none of our results related to the copd-6 have previously been published.

Comment

Response

A minimal data set including the microdata for copd-6 and MicroDL results has already been included in the submission as S4 File. As indicated in the manuscript file, information on family relationships and demographics has not been included in the minimal data file, as doing so would pose a threat to participant confidentiality, which would be unethical. Due to the way that participants were recruited (members of two small agricultural organizations in a small, well-defined geographical area), we deem that pseudonymization is insufficient to ensure that participants cannot be re-identified if the microdata on demographics or family relationships are made public. 

The full dataset is stored in the archives of Aarhus University, Department of Public Health, and we have updated the contact information in the section “Availability of data and materials” so that requests for data access should now be sent to the department (instead of the corresponding author). This will ensure that the data remains available, even if the corresponding author is no longer affiliated with the department.

 

Reviewer: 1

Comment

Considering methods of performing spirometry I have some objections. Investigation team stated 5 efforts on a diagnostic-quality spirometry, and eventually 4 additional blew if participants did not fulfill standard quality criteria, while according to the ATS/ERS guideline it should be at least three, and most 8 maneuvers for acceptable spirometry. On the copd-6 spirometer they measured just three maneuvers, what is an obstacle, because fatigue is expected after too many efforts of forces expiration. Here were 12 blew into devises in some patients, what it much, and put in question quality of results.

Response

We agree that in principle, participant fatigue could have posed a problem for the validity of our findings, due to the number of blows that each participant performed. However, sensitivity analyses stratified by order of testing (copd-6 before or after MicroDL), plus limited to the first three blows with the MicroDL device, gave similar results as the main analysis, indicating that our findings are unlikely to be considerably influenced by patient fatigue. This is discussed in a new paragraph in the “Discussion” section.

Comment

Time frame between two spirometric measurements is not stated on a diagnostic-quality spirometer and on a copd-6 spirometer, what is another question.

Response

We have now added the following sentence in the section “Methods” � “Spirometry testing”:

“Only a few minutes passed between testing with each of the two devices.”

Comment

Also, due to diurnal variation of bronchial muscle tone, it is necessary to put the information of time during the day when the measurements were performed.

Response

In the section “Methods” � “Spirometry testing”, we have added the sentence “Tests were conducted between 7 AM and 5 PM.”

We do not think that diurnal variation in dynamic lung volumes poses a threat to our analyses. As described above, only a few minutes passed between the copd-6 test and the MicroDL test for each participant. Any diurnal effect on FEV1, FVC, and FEV6 should therefore affect the copd-6 and MicroDL devices equally. To keep the manuscript concise, we have chosen not to include discussion of diurnal effects in the paper, but if the editor prefers we will be glad to include it.

Comment

Again, considering cut-off value for non repeatable spirometric result between two best FEV1 and FVC measurements, the chosen value of difference < 0.25 L is too high, due to the ATS/ERS guideline, the cut-off for difference should be < 0.15 L.

Response

We are aware that according to the ATS guidelines, the best and second-best values of FEV1 and FVC should be < 0.15 L. However, using this reproducibility criterion would leave only 120 measurements of FEV1 to be analyzed. Due to the theoretical risk of selection bias when discarding the majority of the measurements, we decided to use the less strict reproducibility criterion of < 0.25 L for our main analysis, which allowed us to analyze 171 measurements instead. A sensitivity analysis limited to the 120 measurements with stricter repeatability gave similar results as the main analysis. This is discussed in a new paragraph in the “Discussion” section.

Comment

Another unusual results is that in healthy, rather young and not obese population (46.6 years and BMI 23.3), only 47% of the investigated subjects fulfilled quality criteria for both spirometry devices. Does it mean that the personal was not enough motivated for the participants, or why – this fact need further explanation.

Response

We have added a new paragraph in the “Discussion” section where we discuss the high number of copd-6 results that were discarded. Since the same technicians were responsible for testing with the copd-6 and MicroDL devices, this difference is unlikely to be due to problems with coaching, as poor coaching would have influenced both copd-6 and MicroDL results. Furthermore, the corresponding author supervised the spirometry technicians during data collection, and it was his impression that the technicians vigorously coached the participants during each blow. Out of the 125 persons whose copd-6 results were excluded, 98 (78%) were excluded because they had performed less than two blows without warnings. It could be speculated that the copd-6 is perhaps too conservative when assessing the quality of individual blows (and labelling them as “unacceptable” by giving a warning), but since the device does not export spirograms for manual review, we cannot investigate this further.

 

Reviewer: 2

Comment

The high drop out in the COPD6 group should be explained in more details and elaborated in the discussion and the conclusion.

Response

As described above, we have added a new paragraph in the “Discussion” section where we discuss the high number of copd-6 results that were discarded.

Comment

The fact that the investigation was done in the population without obstruction should be more accentuated in the conclusion.

Response

We have revised the first sentence of the conclusion so that it now reads as follows:

“Overall, in our relatively healthy study population the copd-6 slightly overestimates FEV1 by 0.04 L, but the bias is flow-dependent.”

Comment

106- less than two or <2 not mixed

Response

Wording in line 106 changed to “less than two” as suggested.

 

Comments for figures

Comment

Response

The figures have now been processed using PACE and re-uploaded.

---

## [Decision Letter · Decision Letter 1]

6 Apr 2021

PONE-D-20-31069R1

The Vitalograph copd-6 mini-spirometer as more than a screening device: Validation of FEV1 in a healthy Ugandan population

PLOS ONE

Dear Dr. Hansen,

Thank you for submitting your manuscript to PLOS ONE. After careful consideration, we feel that it has merit but does not fully meet PLOS ONE’s publication criteria as it currently stands. Therefore, we invite you to submit a revised version of the manuscript that addresses the points raised during the review process.

Please do the changes suggested by the second reviewer as the article theme is not COPD.

We look forward to receiving your revised manuscript.

Kind regards,

Davor Plavec, MD, MSc, PhD, Prof.

Academic Editor

PLOS ONE

Journal Requirements:

Additional Editor Comments (if provided):

Dear Authors, as COPD is not the theme of your research please do the suggested changes by the second reviewer.

Reviewers' comments:

Reviewer's Responses to Questions

**Comments to the Author**

1. If the authors have adequately addressed your comments raised in a previous round of review and you feel that this manuscript is now acceptable for publication, you may indicate that here to bypass the “Comments to the Author” section, enter your conflict of interest statement in the “Confidential to Editor” section, and submit your "Accept" recommendation.

Reviewer #1: All comments have been addressed

Reviewer #2: (No Response)

2. Is the manuscript technically sound, and do the data support the conclusions?

Reviewer #1: Yes

Reviewer #2: No

3. Has the statistical analysis been performed appropriately and rigorously? 

Reviewer #1: Yes

Reviewer #2: I Don't Know

4. Have the authors made all data underlying the findings in their manuscript fully available?

Reviewer #1: Yes

Reviewer #2: Yes

5. Is the manuscript presented in an intelligible fashion and written in standard English?

Reviewer #1: Yes

Reviewer #2: Yes

6. Review Comments to the Author

Reviewer #1: Every line of comments were considered and addressed appropriately and seriously. They have answered and added discussion on every topic rised. I have no additional comments.

Reviewer #2: COPD is diagnosed based on FEV1/FVC ratio. Since in this article only FEV1 accuracy was measured and compared, the title and the whole text should be arranged accordingly. So only what is prooved should be stated. There is no place of mentioning the COPD diagnosis in the context of these data. All references to COPD should be omitted from the introduction, discussion and conclusion or explained how they are connected to FEV1 measuring. So, we can see only the results of FEV1 accuracy in the healty population.

7. PLOS authors have the option to publish the peer review history of their article (what does this mean?). If published, this will include your full peer review and any attached files.

Reviewer #1: **Yes: **Sanja Popović-Grle

Reviewer #2: No

---

## [Author Response · Author response to Decision Letter 1]

20 May 2021

Journal requirements

Comment

Response

We have reviewed our reference list and confirm that is complete and correct. We have not cited any retracted papers. As previously described, parts of table 1 and 2 have been published before and are reproduced in the current paper with permission from the publisher. Since the previous paper has now been published in final form, we have moved its citation text from “Acknowledgements” to “References”.

Additional editor comments

Comment

Dear Authors, as COPD is not the theme of your research please do the suggested changes by the second reviewer.

Response

Done as requested (please see below).

 

Reviewer: 1

Comment

Every line of comments were considered and addressed appropriately and seriously. They have answered and added discussion on every topic rised. I have no additional comments.

Response

Thank you for our comments that have helped us improve the manuscript.

Reviewer: 2

Comment

COPD is diagnosed based on FEV1/FVC ratio. Since in this article only FEV1 accuracy was measured and compared, the title and the whole text should be arranged accordingly. So only what is prooved should be stated. There is no place of mentioning the COPD diagnosis in the context of these data. All references to COPD should be omitted from the introduction, discussion and conclusion or explained how they are connected to FEV1 measuring. So, we can see only the results of FEV1 accuracy in the healty population.

Response

We acknowledge that COPD is diagnosed based on FEV1/FVC ratio and not FEV1 itself. As requested, we have updated the title, the introduction, discussion and conclusions so that the text does not suggest otherwise.

We have kept a brief section on COPD prevalence and diagnosis in the introduction, since we think that it is important background knowledge for the reader. The high burden of disease due to COPD is the reason why our study is relevant: An affordable, precise and accurate instrument for the determination of pulmonary function would have the potential to improve epidemiological research on pulmonary function impairment, especially in resource-constrained settings.

---

## [Decision Letter · Decision Letter 2]

3 Jun 2021

Precision and accuracy of FEV_1_ measurements from the Vitalograph copd-6 mini-spirometer in a healthy Ugandan population

PONE-D-20-31069R2

Dear Dr. Hansen,

We’re pleased to inform you that your manuscript has been judged scientifically suitable for publication and will be formally accepted for publication once it meets all outstanding technical requirements.

Kind regards,

Davor Plavec, MD, MSc, PhD, Prof.

Academic Editor

PLOS ONE

Additional Editor Comments (optional):

Reviewers' comments:

Reviewer's Responses to Questions

**Comments to the Author**

1. If the authors have adequately addressed your comments raised in a previous round of review and you feel that this manuscript is now acceptable for publication, you may indicate that here to bypass the “Comments to the Author” section, enter your conflict of interest statement in the “Confidential to Editor” section, and submit your "Accept" recommendation.

Reviewer #1: All comments have been addressed

2. Is the manuscript technically sound, and do the data support the conclusions?

Reviewer #1: Yes

3. Has the statistical analysis been performed appropriately and rigorously? 

Reviewer #1: Yes

4. Have the authors made all data underlying the findings in their manuscript fully available?

Reviewer #1: Yes

5. Is the manuscript presented in an intelligible fashion and written in standard English?

Reviewer #1: Yes

6. Review Comments to the Author

Reviewer #1: As all lines, and all comments which I have suggested in the previous revision were adequately addressed, from my point of view there is no need for further actions.

7. PLOS authors have the option to publish the peer review history of their article (what does this mean?). If published, this will include your full peer review and any attached files.

Reviewer #1: **Yes: **Sanja Popović-Grle

---

## [Editor Report · Acceptance letter]

18 Jun 2021

PONE-D-20-31069R2 

Precision and accuracy of FEV_1_ measurements from the Vitalograph copd-6 mini-spirometer in a healthy Ugandan population 

Dear Dr. Hansen:

I'm pleased to inform you that your manuscript has been deemed suitable for publication in PLOS ONE. Congratulations! Your manuscript is now with our production department. 

Kind regards, 

on behalf of

Dr. Davor Plavec 

Academic Editor

PLOS ONE